# Robustness Tests for Automatic Machine Translation Metrics with Adversarial Attacks

**Yichen Huang**
MBZUAI
yichen.huang@mbzuai.ac.ae

**Timothy Baldwin**
MBZUAI
The University of Melbourne
timothy.baldwin@mbzuai.ac.ae

## Abstract

We investigate MT evaluation metric performance on adversarially-synthesized texts, to shed light on metric robustness. We experiment with word- and character-level attacks on three popular machine translation metrics: BERTScore, BLEURT, and COMET. Our human experiments validate that automatic metrics tend to overpenalize adversarially-degraded translations. We also identify inconsistencies in BERTScore ratings, where it judges the original sentence and the adversarially-degraded one as similar, while judging the degraded translation as notably worse than the original with respect to the reference. We identify patterns of brittleness that motivate more robust metric development.

## 1 Introduction

Automatic evaluation metrics are a key tool in modern-day machine translation (MT) as a quick and inexpensive proxy for human judgements. The most common and direct means to evaluate an automatic metric is to test its correlation with human judgements on outputs of MT systems. However, as such metrics are commonly used to inform the development of new MT systems and even used as training and decoding objectives (Wieting et al., 2019; Fernandes et al., 2022), it is inevitable for them to be applied to out-of-distribution texts that do not frequently occur in existing system outputs. The rapid advancement of MT systems and metrics, as well as the prospect of incorporating MT metrics in the training and generation process, motivates investigation into MT metric robustness.

In this work, we examine textual adversarial attacks (TAAs) as a means to synthesize challenging translation hypotheses where automatic metrics systematically underperform. We experiment with word- (Li et al., 2021; Jia et al., 2019; Feng et al., 2018) and character-level (Gao et al., 2018) attacks on three popular, high-performing automatic

COMET(original) → COMET(perturbed):
2.081 → -0.363 (-2.444)

Human(orignal) → Human(perturbed):
1.225 → 0.274 (-0.951)

Reference:
Notice: I don't have an oven, just a large pot.

Original translation:
Note: I don't have an oven, just a large pot.

Perturbed translation:
Note: I don't have an oven, just a large casserole.

(a)

BERTScore(original, original) → BERTScore(perturbed, original):
2.127 → 1.860 (-0.268)

BERTScore(original, reference) → BERTScore(perturbed, reference):
0.325 → -0.077 (-0.402)

Reference:
There are many Latin American elements in its tempo and melodies.

Original translation:
There are a lot of Latin American ingredients in your Tempo and melody.

Perturbed translation:
There are a lot of good Latin American ingredients in your Tempo and melody.

(b)

Figure 1: (a) The metric overpenalizes the perturbed translation when compared with human ratings. (b) The metric is self-inconsistent as it judges the original and perturbed translations to be similar (BERTScore(original, original) → (BERTScore(perturbed, original)) while judging the perturbed sentence as a worse translation (BERTScore(original, reference) → BERTScore(perturbed, reference)). All ratings are normalized.

MT metrics: BERTScore (Zhang et al., 2020), BLEURT (Sellam et al., 2020), and COMET (Rei et al., 2020). We construct situations where the metrics disproportionately penalize adversarially-degraded translations. To validate such situations, we collect a large set of human ratings on both original and adversarially-degraded translations. As BERTScore can also be seen as a measure of semantic similarity between any two sentences, we also explore another scenario of inconsistency where BERTScore judges the original and adversarially-

perturbed translations as similar while judging the perturbed translation as notably worse than the original one with regard to the reference translation. Examples are shown in Figure 1.

We identify mask-filling and word substitution as effective means to generate perturbed translations where BERTScore, BLEURT, and COMET over-penalize degraded translations and BERTScore is self-inconsistent. In particular, BLEURT and COMET are more susceptible to perturbations in data with higher-quality translations. Our findings serve as a basis for developing more robust automatic MT metrics.[1]

## 2 Methods

### 2.1 Formulation

Most TAA methods probe for the overreaction of the victim model $f$ (Wang et al., 2022). Given the original text $x$ and associated label $y$, the methods generate a bounded perturbed $x'$ with label $y'$. The perturbation is assumed to be label-preserving (i.e. $y' = y$). Robust behavior would be $f(x') = y$ for classification or $f(x') \approx y$ for regression, and the attack is considered successful iff $f(x')$ is notably different to $y$. The label-preserving assumption is usually enforced by a set of constraints.

In our task, given the original translation $x$ and metric rating $y$, we aim to generate a perturbed text $x'$ that misleads the metric $f$ such that $f(x')$ is notably different from $y$. The label-preserving assumption amounts to equivalence in meaning and fluency, which is commonly enforced through sentence embedding distance (Li et al., 2021) and perplexity (Jia et al., 2019; Alzantot et al., 2018). However, semantic equivalence can clearly not be adequately enforced in our case: the MT metric can roughly be seen as a model-based measure of semantic similarity, similar to the sentence embedding model enforcing the semantic constraint. When we have a "successful" attack where $f(x')$ is notably different from $y$, we cannot be certain whether it is because we have a faulty metric (where the ground truth $y'$ is close to $y$ but $f(x')$ is notably different from $y'$) or a faulty constraint (where the perturbed $x'$ is semantically different from $x$ and thus $y'$ should be different from $y$).

We explore two approaches with regard to this issue. Firstly, we experiment with forgoing the semantic constraint and searching for $x'$ such that $f(x')$ is notably lower than $y$ with a minimal number of perturbations under only the fluency constraint. The intuition is that when the number of perturbations is small, humans are likely to rate the extent of degradation as less significant than the automatic metric. To validate whether the assumption holds, we collect continuous human ratings on meaning preservation against the reference translation following Graham et al. (2013, 2014, 2017) and compare the extent of degradation as judged by humans and that as judged by the metrics. We focus on meaning preservation as it is aligned with the training objectives of BLEURT and COMET. We describe further details in Appendix A.

Secondly, we investigate a scenario where BERTScore is self-inconsistent by using itself as a semantic similarity constraint. As BERTScore can be seen as a generic distance metric of semantic similarity, we can use it to measure the distance between the original and the perturbed translations, the original translation and the reference, as well as the perturbed translation and the reference. When the original and perturbed translations are measured as similar, the robust behavior would be for them to have similar ratings with regard to the reference. We search for violations against this where BERTScore(perturbed, reference) is notably smaller than BERTScore(original, reference), but BERTScore(perturbed, original) is close to BERTScore(original, original), which we use as a maximum score of similarity.[2]

### 2.2 Adversarial Attack Setup

We use the German-to-English system outputs from WMT 12, 17, and 22 (Callison-Burch et al., 2012; Bojar et al., 2017; Kocmi et al., 2022), and randomly select 500 sentences for each system for each year, totalling 19K (source, translation, reference) tuples. For the sake of efficiency, we use MT outputs whose associated references are longer than 10 words. We normalize each metric such that their outputs on this dataset have a mean of 0 and a standard deviation of 1. When probing for overpenalization, we consider three widely-used metrics: BERTScore (Zhang et al., 2020), BLEURT (Sellam et al., 2020), and COMET (Rei et al., 2020). We constrain the perturbed sentence to have an increase in perplexity of no more than

---

[1]Code and data are available at https://github.com/i-need-sleep/eval_attack.

[2]We attempted a similar setup with BLEURT, using a symmetric variant of BLEURT as the semantic constraint, but preliminary experiments returned no successful attacks. We discuss further details in Appendix C.

| Method | BERTScore | | | BLEURT | | | COMET | | |
|---|---|---|---|---|---|---|---|---|---|
| | WMT 12 | WMT 17 | WMT 22 | WMT 12 | WMT 17 | WMT 22 | WMT 12 | WMT 17 | WMT 22 |
| CLARE | 98.56% (7885) | 99.44% (5469) | 99.38% (5466) | 98.74% (7899) | 99.91% (5495) | 99.62% (5479) | 20.28% (1622) | 20.78% (1143) | 20.73% (1140) |
| Faster Genetic | 75.99% (6079) | 75.67% (4162) | 70.84% (3896) | 76.21% (6097) | 80.42% (4423) | 75.09% (4130) | 11.74% (939) | 12.25% (674) | 11.80% (649) |
| Input Reduction | 40.01% (3201) | 38.16% (2099) | 43.76% (2407) | 31.46% (2517) | 30.96% (1703) | 34.24% (1883) | 50.01% (4001) | 50.49% (2777) | 52.82% (2905) |
| DeepWordBug | 10.93% (874) | 6.75% (371) | 9.09% (500) | 12.32% (986) | 7.07% (389) | 6.55% (360) | 17.95% (1436) | 10.75% (591) | 9.62% (529) |

Table 1: The percentages and numbers of perturbations fitting our criteria for each metric for each year. The WMT 12, 17, and 22 splits ontain 8K, 5.5K and 5.5K original system outputs, respectively.

10 as measured by GPT-2 (Radford et al., 2019), and search for cases where the perturbed translation causes a decrease of more than 1 standard deviation in the metric rating. When probing for self-inconsistency with BERTScore, we constrain the difference between BERTScore(perturbed, original) and BERTScore(original, original) to be less than 0.3 after normalization, and search for cases where the perturbed translation causes a decrease of more than 0.4 in BERTScore.

For both setups, we apply a range of black-box search methods to generate perturbations, including word-level attacks (CLARE (Li et al., 2021), the Faster Alzantot Genetic Algorithm (Jia et al., 2019), Input Reduction (Feng et al., 2018)) and character-level attacks (DeepWordBug (Gao et al., 2018)). CLARE applies word replacements, insertions, and merges by mask-filling, the Faster Alzantot Genetric Algorithm applies word substitutions, Input Reduction applies word deletions, and DeepWordBug applies character swapping, substitution, deletion, and insertion. We further describe these methods in Appendix B.

## 3 Results

### 3.1 Probing for Overpenalization

We generate a total of 102,176 perturbed translations fitting our criteria. The breakdown across the search methods, metrics, and years is shown in Table 1. All three metrics seem insensitive to character-level perturbations, with DeepWordBug returning a small number of eligible perturbations for each year. The more sophisticated CLARE and Faster Alzantot Genetic Algorithm returns a larger ratio of eligible perturbations for BERTScore and BLEURT. On the contrary, COMET appears more sensitive to word deletions, with Input Reduction returning eligible perturbations for more than 50% of the system outputs. The ratio of eligible perturbations fluctuates only slightly for different years.

We collect human ratings for a balanced subset of eligible perturbated translations and corresponding original translations. We aggregate the normal-ized ratings across annotators, resulting in 2,800 qualifying ratings respectively for original and perturbed sentences. The Pearson $r$ correlations with human ratings are shown in Figure 2. We observe that the attacks lead to worsened correlations in most cases, with CLARE and the Faster Alzantot Genetic Algorithm leading to bigger degradations, suggesting mask-filling and word substitution as effective means of attack. All three metrics are particularly susceptible to perturbations on the WMT 22 data where the original translations are of higher quality. Both CLARE and the Faster Alzantot Genetic Algorithm lead to degradations of over 0.2 in Pearson correlations for BLEURT and over 0.4 for COMET. This is likely because BLEURT and COMET are trained on data from previous years and cannot easily generalize to higher-quality translations with minor modifications.

To investigate the cause of the reduced correlations, we compare the degradation of translation quality as measured by the metrics and as judged by humans. Results are shown in Figure 5. We observe that, in most cases, the metrics assign higher differences between the original and perturbed translations. To quantify this observation, we perform a one-sided Wilcoxon rank-sum test on the subsets of the data corresponding to the 36 combinations of metrics, years, and attack methods. Under a significance level of $p < 0.05$, for 25 out of the 36 combinations, the degradation as measured by the metrics is significantly larger than that as measured by humans. This confirms our assumption of overpenalization. We also find that in most cases, the metrics penalize different perturbation instances more consistently than humans. As an exception, BLEURT and COMET are significantly more inconsistent when measuring CLARE-generated degradations for WMT 22. This, again, suggests vulnerability against perturbed, high-quality translations outside the models' training sets.

We also investigate the influence of sentence length on overpenalization. Changing a word in a short sentence may result in a larger score differ-

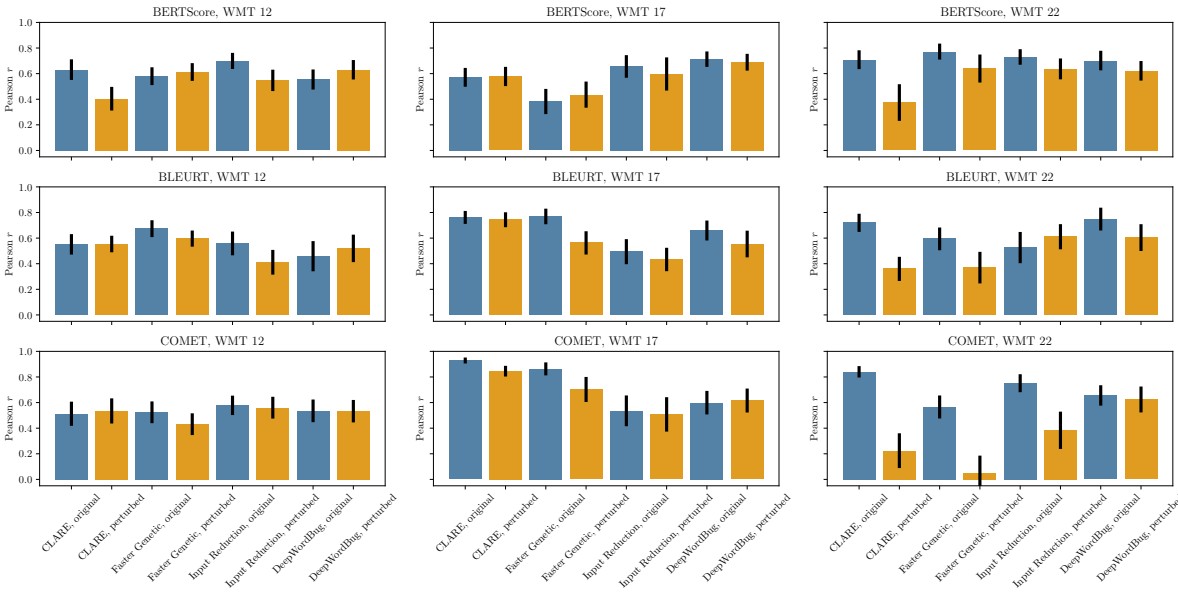

Figure 2: The Pearson correlation ($r$) with human ratings for different metrics, years, and attack methods on original and perturbed translations. The error bars show the standard error as computed through bootstraping with 10K resamples.

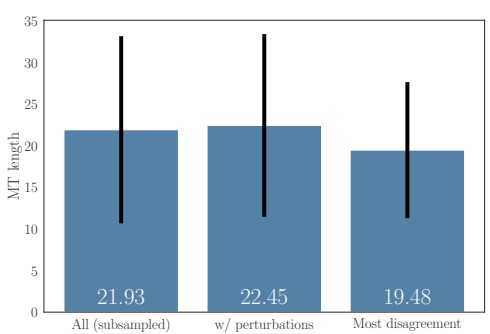

Figure 3: MT output length in the number of words for: (1) the 19K MT outputs we used to generate the perturbations (subsampled to be balanced across the years); (2) all MT outputs leading to eligible perturbations; and (3) the 500 MT outputs where humans and metrics disagree the most on the degree of penalization.

ence, and this difference might be different between human and metric scores. We compare the length of MT outputs (in the number of words) before perturbation for the following subsets: (1) the 19K MT outputs we used to generate the perturbations (subsampled to be balanced across the years); (2) all MT outputs leading to eligible perturbations; and (3) the 500 MT outputs where humans and metrics disagree the most on the degree of penalization. Statistics are shown in Figure 3. Using a one-sided Wilcoxon rank-sum test with $p < 0.05$,

| Method | WMT 12 | WMT 17 | WMT 22 |
|---|---|---|---|
| CLARE | 3.68% (49) | 5.78% (77) | 2.55% (34) |
| Faster Genetic | 0.83% (11) | 1.58% (21) | 0.98% (13) |
| Input Reduction | 1.28% (17) | 0.23% (3) | 0.38% (5) |

Table 2: Percentages and numbers of successful attacks on BERTScore for self-inconsistency on 4K randomly-sampled sentences. DeepWordBug returns no successful attacks.

we find that (3) is smaller than (1) and (2) at a level of statistical significance. This suggests that shorter MT outputs lead to more severe over-penalization. However, the difference in sentence lengths is small and does not fully explain the different degrees of penalization.

## 3.2 Probing for Self-Inconsistency

We randomly select a total of 4K system outputs balanced across years and systems, and search for perturbations fitting our criterion of self-inconsistency. Results are shown in Table 2. While all attack methods return a small number of successful attacks, we observe a trend that CLARE and the Faster Alzantot Genetic Algorithm have a higher success rate. This, again, suggests the effectiveness of mask-filling and word substitution as attack methods.

### 3.3 Implications of this Work

The immediate implication of this work is to augment training of learned metrics such as BLEURT and COMET with the data generated in this work, and experiment with incorporating automatically-generated synthetic data based on mask-filling and word substitution.

### 4 Related Work

Classical MT metrics such as BLEU (Papineni et al., 2002) and ROUGE (Lin, 2004) have been shown to correlate poorly with human judgements (Mathur et al., 2020; Kocmi et al., 2022), motivating the development of model-based metrics. Supervised metrics such as BLEURT (Sellam et al., 2020) and COMET (Rei et al., 2020) are trained to mimic human ratings as a regression task. Non-learnable metrics such as BERTScore (Zhang et al., 2020), XMoverDistance (Zhao et al., 2020), and UScore (Belouadi and Eger, 2023) do not rely on human ratings and instead leverage the embeddings of the source, reference, and translation. Whereas more recent and higher-performing metrics exist, we focus our investigation on BERTScore, BLEURT, and COMET as they are most commonly used and easily adapted to other domains such as text simplification (Maddela et al., 2023).

Wang et al. (2022) defines robustness as performance on unseen test distributions. Such distributions can occur naturally (Hendrycks et al., 2021) or be constructed adversarially, and robustness failures are usually identified through human priors and error analysis. Alves et al. (2022) and Chen et al. (2022) use hand-crafted types of perturbations to create challenge sets where MT metrics underpenalize the perturbed translations. Yan et al. (2023) use minumum risk training (Shen et al., 2016) to optimize directly for higher metric scores, resulting in a set of translations with overestimated scores. This work complements previous works by investigating overpenalization, which is a very different behavior to overestimation. We use adversarial attacks targeted at each metric that are not limited to pre-defined categories, which allows us to discover particular failure cases specific to each metric. In addition, we consider the same set of metrics on different years of WMT data. This allows us to draw connections between adversarial robustness and the quality of MT system outputs, and whether the MT system outputs are used when training the metric.

### 5 Conclusion

We apply word- and character-level adversarial attacks and probe for overpenalization with BERTScore, BLEURT, and COMET, and for self-inconsistencies with BERTScore. We observe that mask-filling and word substitution are more effective at generativing challenging cases, and that BLEURT and COMET are more susceptible to perturbation of high-quality translations.

Our findings motivate more sophisticated data augmentation and training methods to achieve greater metric robustness. In particular, our formulation of self-consistency requires no validation against human ratings and can be applied to other embedding-based metrics (Zhao et al., 2020; Reimers and Gurevych, 2020; Belouadi and Eger, 2023) as a regularization term. We leave this to future work.

### 6 Limitations

We use the high-resource German-to-English subset of the WMT datasets, which mainly focuses on the news domain. How readily our results translate across to other language pairs, translation systems, metrics, or domains requires further investigation. We experiment with only word- and character-level attacks, but other methods exist that generate sentence-level (Ross et al., 2022) or multi-level (Chen et al., 2021) attacks. We leave a more comprehensive study of attack methods to future work.

### 7 Ethics Statement

The human ratings are collected from fluent English speakers contracted through a work-sharing company. The annotators are paid fairly according to local standards. Prior to annotation, we informed the annotators of the purpose of the collected data and provided relevant training. We ensured that the annotators had a prompt means of contacting us throughout the annotation process.

### 8 Acknowledgements

We extend our gratitude to Teresa Lynn for her assistance with experiment design and data collection. We thank Daniel Chin and Yuchen Wang for their feedback on the initial draft of this work. Finally, we thank the anonymous reviewers for their helpful feedback and suggestions.

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

## A  Annotation

To validate whether the perturbed sentences lead to degraded metric performance, we collected human ratings for a subset of perturbed translations and corresponding original system outputs. We largely follow the DA protocol (Graham et al., 2013, 2014, 2017) and task the annotators to rate on a continuous scale to what extent the meaning of the reference is expressed in the translations. We focus on meaning preservation as it is aligned with the training objectives of BLEURT and COMET. We take a more conservative stance by displaying the original and perturbed sentences in parallel and highlighting the perturbed words. Our intuition is that the annotators are more inclined to exaggerate the quality differences between the original and perturbed translations under this setup. An example of the annotation interface is shown in Figure 4. We randomise the order of the original and perturbed sentences such that it is not immediately clear to the annotator which is which.

For each human intelligence task (HIT) of 100 (reference, original, perturbed) tuples, we use the ratings from 70 tuples and include 30 tuples as control items. 15 of the control items are duplicates from the 70 tuples, and the other 15 contain degraded original and perturbed translations from the 70 tuples, where we randomly drop four words. We use the Wilcoxon rank sum test to ensure that the score differences from the duplicated pairs are smaller than that of the degraded pairs. We reject HITs where the $p$ value is larger than 0.05.

We collect ratings from 10 fluent English speakers[3] contracted through a work-sharing company. We conduct training sessions where we describe the task and annotation interface prior to the annotation process. In total, we collected 268 HITs, with 177 ($66.04\%$) HITs passing quality control. The ratio is higher than those reported by Graham et al. (2017) as we work with trained annotators. We ob-

---

[3]The annotation team is lead by two native English speakers. Each of the remaining members either completed undergraduate education in English or have spent years living in the UK, and have the equivalent of C2 proficiency in English.

tain the $z$-scores by normalizing annotations from the same annotator, and aggregate the ratings for the same translation by averaging. We use tuples with at least three annotations, resulting in 10,080 annotations for 2,800 tuples. The annotated data is balanced for the different metrics, years, and search methods.

## B  Implementation Details

We use the Huggingface `Evaluate`[4] implementations of BERTScore, BLEURT, and COMET. For BERTScore, we use `roberta-large` as the underlying model and use F1 score as the metric output. For BLEURT, we use the improved `bleurt-20-d12` checkpoint introduced by Pu et al. (2021). For COMET, we use the `wmt20-comet-da` checkpoint.

We consider four search methods for adversarial attacks: CLARE, the Faster Alzantot Genetic Algorithm, Input Reduction, and DeepWordBug. CLARE iteratively applies contextualized word-level replacements, insertions, and merges by masking and bounded infilling, with each perturbation greedily selected by the impact on the victim. The Faster Alzantot Genetic Algorithm modifies the genetic algorithm proposed by Alzantot et al. (2018), and iteratively searches for word replacements that are close in a counter-fitted embedding space (Mrkšić et al., 2016). Input Reduction iteratively removes the least important word based on its influence on the victim's output. DeepWordBug iteratively applies a heuristic set of scores to determine the word to perturb, and applies character level swapping, substitution, deletion, and insertion.

We use the `TextAttack` (Morris et al., 2020) implementations of the adversarial attacks. For CLARE, we modify the default implementation by removing the sentence similarity constraint and using beam search with width 2 and a maximum of 10 iterations when probing for overpenalization, and with width 5 and a maximum of 15 iterations when probing for self-inconsistency. For the Faster Alzantot Genetic Algorithm, we modify the implementation by changing the LM constraint and using a population size of 30 and a maximum of 15 iterations when probing for overpenalization, and a population size of 60 and a maximum of 40 iterations when probing for self-inconsistency. We follow the default implementation otherwise. For the GPT-2 perplexity constraint, we use the

---

[4]https://huggingface.co/docs/evaluate/index

Figure 4: A screenshot of the annotation interface. The slider is initialized at the middle position. The annotator must interact with the slider before proceeding to the next annotation instance and cannot revisit completed annotations.

Huggingface `Evaluate` implementation.

## C Probing for Self-Inconsistency with BLEURT

Our formulation of self-consistency does not immediately apply to BLEURT as it distinguishes between the hypothesis and the reference and thus cannot be seen as a distance metric. We instead experiment with a symmetric variant of BLERUT as the semantic constraint. Given BLEURT(hypothesis, reference), we define symmetric BLEURT as the average between BLEURT(original, perturbed) and BLEURT(perturbed, original). We constrain the difference between BLEURT(original, original) and this symmetric measure to be smaller than 0.3. We search for perturbed translations such that the difference between BLERUT(original, reference) and BLERUT(perturbed, reference) is larger than 0.4.

None of the search methods returns successful attacks for our preliminary experiments with 1K randomly sampled translations. We find that BLEURT overpenalizes translations with low-quality references, i.e. BLEURT(original, perturbed) is significantly smaller than BLERUT(perturbed, original).

This makes it difficult to find perturbations satisfying the semantic constraint.

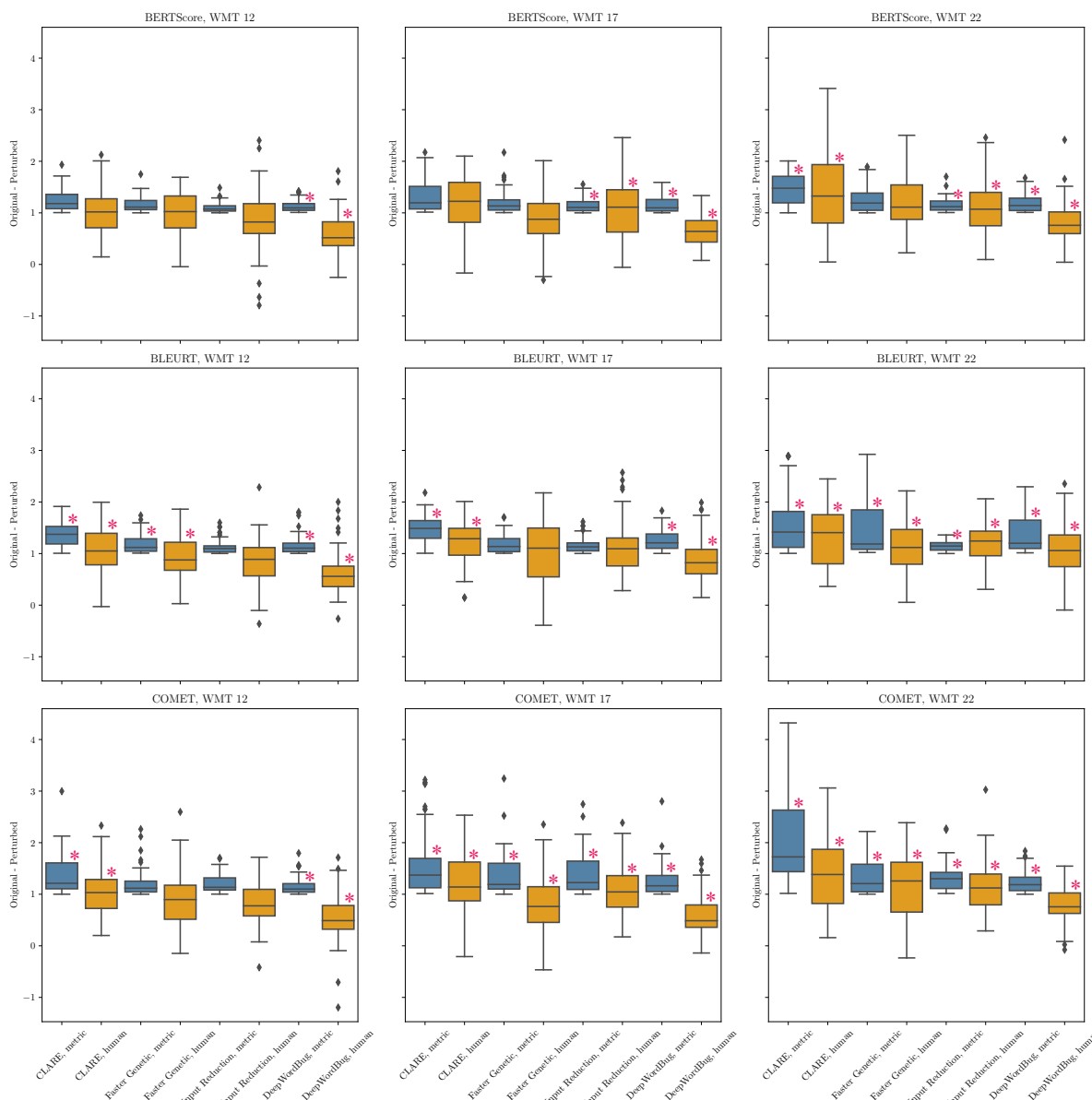

Figure 5: The quality difference between the original and perturbed translations as measured by the metrics and humans. Bars marked with red asterisks are the cases where the degradation for metrics is significantly larger ($p < 0.05$) than that for humans.