# OpenReview forum: "Robustness Tests for Automatic Machine Translation Metrics with Adversarial Attacks"
_EMNLP/2023/Conference — EMNLP 2023 Findings_

### Official Review · Reviewer_JiPk · 2023-07-26

**Soundness:** 3

**Excitement:**

3: Ambivalent: It has merits (e.g., it reports state-of-the-art results, the idea is nice), but there are key weaknesses (e.g., it describes incremental work), and it can significantly benefit from another round of revision. However, I won't object to accepting it if my co-reviewers champion it.

**Paper Topic And Main Contributions:**

This paper probes the robustness of the current neural MT metrics (i.e., BERTScore, BLEURT, COMET) via adversarial attacks. The authors collect lots of human judgments for comparison and finally conclude that these metrics have overpenalization problem and BERTScore performs inconsistently on perturbed translations. Overall, the findings in this paper are somewhat instructive for the development of the robustness of current neural metrics but some statements still need to be clearer.

**Questions For The Authors:**

- I don't really understand the results in Table 2. There are only at most 1~2% successful attacks for the self-inconsistency problem of BERTScore. Does this mean that the self-inconsistency problem is actually not severe at all for BERTScore and the conclusion doesn't hold?

**Reasons To Accept:**

- This paper aims at an important problem in NMT, i.e., the robustness of the current neural metrics for NMT, and conducts comprehensive analyses on three prevalent metrics.
- The findings in this paper are interesting and may motivate future work on enhancing the robustness of the neural metrics for NMT.

**Reasons To Reject:**

- The two differences between BERTScore results in Figure 1(b) seem incorrect, e.g., $1.860-2.127=-0.268?$ and $-0.077-0.325=-2.444?$
- The writing of the paper needs to be improved. Some descriptions of the methods and details in the results are somewhat difficult to follow.
- As the authors stated, the analyses are limited to De-En. It would be more convincing with more results on other language pairs and domains.

**Reproducibility:**

2: Would be hard pressed to reproduce the results. The contribution depends on data that are simply not available outside the author's institution or consortium; not enough details are provided.

**Reviewer Confidence:**

4: Quite sure. I tried to check the important points carefully. It's unlikely, though conceivable, that I missed something that should affect my ratings.

---

> ### Author Rebuttal · Authors · 2023-08-29
>
> Thank you for the insightful feedback and questions. We respond to the questions and concerns in a breakdown format.
>
> > The two differences between BERTScore results in Figure 1(b) seem incorrect.
>
> The difference on the second row is indeed a mistake on our side. The second row should be BERTScore(original, reference) -> BERTScore(perturbed, reference): 0.325 -> -0.077(-0.402). The numbers on the first row should be correct, with rounding to 1.860 and 2.127.
>
> > As the authors stated, the analyses are limited to De-En. It would be more convincing with more results on other language pairs and domains.
>
> We agree that experiments on more language pairs would support more general conclusions, but suggest that such extensive investigations fall outside the scope of our short paper. We point out that it is not uncommon for similar works to include only one language pair (e.g. one of the suggested papers (Chen et al., WMT 2022) also only considers one language pair). As such, we leave investigation of different language pairs and domains to future work.
>
> > I don't really understand the results in Table 2. There are only at most 1~2% successful attacks for the self-inconsistency problem of BERTScore. Does this mean that the self-inconsistency problem is actually not severe at all for BERTScore and the conclusion doesn't hold?
>
> It is true that self-consistency issues only occur for a small number of cases for BERTScore. However, we argue that despite their infrequent occurrence, such failure cases can still cause issues in real-world applications and likely cannot be detected using other methods. In addition, the trend is clear that BERTScore is more likely to be self-inconsistent under mask-filling and word-substitution attacks, which is in agreement with our conclusions. We also stress that this method has no human cost and is easily applicable to other embedding-based metrics, such as Word Mover’s Distance.
>
> Finally, we thank you for raising concerns about the writing of the paper and the reproducibility of the work. May we ask you to elaborate on these issues to help inform further revisions?
>
> Reference:
>
> [Exploring Robustness of Machine Translation Metrics: A Study of Twenty-Two Automatic Metrics in the WMT22 Metric Task](https://aclanthology.org/2022.wmt-1.46) (Chen et al., WMT 2022)

---

### Official Review · Reviewer_1kEg · 2023-08-02

**Typos Grammar Style And Presentation Improvements:** line23
**Soundness:** 3

**Excitement:**

3: Ambivalent: It has merits (e.g., it reports state-of-the-art results, the idea is nice), but there are key weaknesses (e.g., it describes incremental work), and it can significantly benefit from another round of revision. However, I won't object to accepting it if my co-reviewers champion it.

**Paper Topic And Main Contributions:**

This paper investigates the robustness of three prevailing machine translation metrics on sentences with adversarial perturbations. It comes up with two conclusions that 1) these metrics tend to over-penalize adversarially degraded translations; 2) they are inconsistent when evaluating different sentences, through experimenting with several adversarial attack methods on German-English translation dataset.

**Questions For The Authors:**

A.. Regarding Figure 1(a)
* Could this phenomenon be affected by the length of the input sentence?

B. Regarding Figure 1(b)
* What does it mean by calculating Bert (original, original)?
* The calculation in the second row seems to be incorrect?

line152 C. What's the threshold for COMET and BLEURT?

**Reasons To Accept:**

1. The investigated problem is interesting, and I agree that some more convinced metrics should be applied to investigate the robustness of MT systems.
2. The experiments lead to some insightful findings.

**Reasons To Reject:**

1. The evaluation setting has some problems.
* The length of the sentence may affect the evaluation results, i.e., change a word in a short sentence may lead to large difference in the metric score. Therefore, a standardization term may be needed when calculate the differences between metric scores.
* Some points in this paper is not clear enough (see the questions below)



**Reproducibility:**

4: Could mostly reproduce the results, but there may be some variation because of sample variance or minor variations in their interpretation of the protocol or method.

**Reviewer Confidence:**

4: Quite sure. I tried to check the important points carefully. It's unlikely, though conceivable, that I missed something that should affect my ratings.

---

> ### Author Rebuttal · Authors · 2023-08-29
>
> Thank you for the insightful feedback and questions. We respond to the questions and concerns in a breakdown format.
>
> >The length of the sentence may affect the evaluation results, i.e., change a word in a short sentence may lead to large difference in the metric score. Therefore, a standardization term may be needed when calculate the differences between metric scores.
>
> We agree that the length of the sentence might be an influencing factor for the metric scores, but the same is true for human ratings: changing a word in a short sentence will likely significantly alter its meaning and greatly alter the human ratings for meaning preservation. It is still a fair comparison when we compare the degree of penalization as given by humans and metrics.
>
> To investigate the influence of sentence length, we report the statistics of the length of MT outputs before perturbation (in the number of words). Note that for the sake of efficiency, we used MT outputs whose associated references are longer than 10 words when collecting (3).
>
> 1. The 500 MT outputs where humans and metrics disagree the most on the degree of penalization: Mean: 19.482. Standard deviation: 8.179.
> 2. All MT outputs leading to eligible perturbations: Mean: 22.452. Standard deviation: 10.979.
> 3. The 19K MT outputs we used to generate the perturbations (subsampled to be balanced across the years): Mean: 21.807. Standard deviation: 11.138.
>
> Should shorter MT outputs lead to more severe over-penalization, (1) should be notably smaller than (2) and (3). Using a one-sided Mann-Whitney U test with a significance level of 0.05, we found that it is indeed statistically significant that (1) is smaller than (2) and (3). However, the difference in sentence lengths is small and cannot fully explain the different degrees of penalization. We do however acknowledge that the potential impact of sentence length deserves further investigation, and will discuss this in the paper.
>
> > Regarding Figure 1(b), what does it mean by calculating Bert (original, original)?
>
> We calculate BERTScore(original, original) by using the same original sentence as both the hypothesis and reference for BERTScore, resulting in a score of 1 before normalization. This is meant to establish a maximum score of similarity as measured by the metric.
>
> > Regarding Figure 1(b), the calculation in the second row seems to be incorrect?
>
> This is indeed our mistake. The second row should be BERTScore(original, reference) -> BERTScore(perturbed, reference): 0.325 -> -0.077(-0.402).
>
> > line152 C. What's the threshold for COMET and BLEURT?
>
> Our method of probing for self-inconsistency does not immediately apply to COMET and BLEURT as they are not symmetric measures (e.g. BLEURT(hypothesis, reference) is different from BLEURT(reference, hypothesis)). As described in Appendix C, we experimented with a symmetric variant of BLEURT under the same threshold as BERTScore.

---

### Official Review · Reviewer_cxZu · 2023-08-05

**Soundness:** 3

**Excitement:**

3: Ambivalent: It has merits (e.g., it reports state-of-the-art results, the idea is nice), but there are key weaknesses (e.g., it describes incremental work), and it can significantly benefit from another round of revision. However, I won't object to accepting it if my co-reviewers champion it.

**Missing References:**

- https://aclanthology.org/2022.wmt-1.43.pdf
- https://aclanthology.org/2022.wmt-1.46.pdf

**Paper Topic And Main Contributions:**

This paper studies machine translation metrics' robustness to various adversarial attacks. Ideally, we would like to find adversarial attacks that (1) do not change the semantics of the translation and (2) significantly degrade the MT metrics' score. However, (1) is hard to enforce, so the paper proposed two approaches: (1) perturb minimally, then confirm by the human evaluation; (2) for BERTScore, use it to evaluate the semantic similarity between original/perturbed translation, then check if it's consistent with the observed change in metric scores before/after perturbation.

Results show that (1) yields a large number of successful attacks that degrades metric score's correlation with human judgments most of the time, but (2) only returned a small number of successful attacks.

**Reasons To Accept:**

1. MT metric's robustness is an important topic to address.
2. The paper demonstrated a successful path toward generating adversarial attacks for MT metrics. The resulting attack and human evaluation, if released, will be a great resource for improving metric models.
3. The experiment, especially the human evaluation, is solid and carefully designed.

**Reasons To Reject:**

1. I'm not convinced that the over-penalization phenomenon described in L206-210 is real -- in Figure 4, most of the differences between humans & metrics are well within the confidence interval. Even the example in Figure 1 is not very convincing -- the humans are not instructed to give negative scores, so this may just be an effect of different scales.
2. When compared to some (uncited) prior work on MT metric robustness (e.g. https://aclanthology.org/2022.wmt-1.43.pdf, https://aclanthology.org/2022.wmt-1.46.pdf), the novelty of this work is incremental.
3. All the experiments are done only on one language pair (de-en).

**Reproducibility:**

4: Could mostly reproduce the results, but there may be some variation because of sample variance or minor variations in their interpretation of the protocol or method.

**Reviewer Confidence:**

4: Quite sure. I tried to check the important points carefully. It's unlikely, though conceivable, that I missed something that should affect my ratings.

---

> ### Author Rebuttal · Authors · 2023-08-29
>
> Thank you for your insightful feedback. In the following comments, we address the raised concerns in a breakdown format.
>
> > I'm not convinced that the over-penalization phenomenon described in L206-210 is real -- in Figure 4, most of the differences between humans & metrics are well within the confidence interval.
>
> We run a Wilcoxon rank-sum test on the degree of degradation as measured by the metrics vs.  humans, with the null hypothesis being that the differences are drawn from the same distribution and the alternative hypothesis that the differences between metric scores are greater. On all data reported in Figure 4, the test results in a very low p-value of $1.65 \times 10^{-38}$, suggesting that overall, the difference is highly significant and the over-penalization phenomenon is real. We also run the test on subsets of the data corresponding to different combinations of metrics, years and attack methods under a significance level of 0.05. Out of the 36 combinations, the 11 cases where the null hypothesis is not rejected are as follows.
>
> * BERTScore
>   * 2012: Clare, faster genetic, input reduction
>   * 2017: Clare, faster genetic
>   * 2022: Faster genetic
> * BLEURT
>   * 2012: Input reduction
>   * 2017: Faster genetic, input reduction
> * COMET
>   * 2012: faster genetic
>
> In the other 25 cases, the over-penalization is statistically significant. These include cases on WMT 22, and cases with CLARE and the Faster Alzantot Genetic Algorithm, where we observed the most significant degradation in the performance of the metrics. This supports our findings and will be included in the paper.
>
> > Even the example in Figure 1 is not very convincing -- the humans are not instructed to give negative scores, so this may just be an effect of different scales.
>
> The human ratings and metric scores in Figure 1 are normalized to have a mean of 0 and standard deviation of 1 and are of the same scale.
>
> > When compared to some (uncited) prior work on MT metric robustness (e.g. https://aclanthology.org/2022.wmt-1.43.pdf, https://aclanthology.org/2022.wmt-1.46.pdf), the novelty of this work is incremental.
>
> Thank you for the suggested papers. We agree that they are relevant and will include them in the revised version. We point out the following important differences with our work:
>
> * The referred works test MT metrics on under-penalization. We complement their findings by investigating over-penalization, which is a very different behaviour.
> * The referred works are based on hand-crafted types of perturbations (named entities, numbers, negations, insertions and deletions for Alves et al. (WMT 2022), and numbers, dates/times, named entities/terminology, unit and affirmation/negations for Chen et al., (WMT 2022)) and do not take into account the response of the metric when generating perturbations. We use adversarial attacks targeted at each metric that are not limited to pre-defined categories (e.g. neither of the referred challenge sets includes free-form substitution). This allows us to discover particular failure cases specific to each metric.
> * The referred works consider different metrics on the same MT dataset, while we consider the same set of metrics on different years of WMT data. This allows us to draw connections between adversarial robustness and the quality of MT system outputs, and whether the MT system outputs are used when training the metric.
>
> > All the experiments are done only on one language pair (de-en).
>
> We agree that experiments on more language pairs would support more general conclusions, but suggest that such extensive investigations fall outside the scope of our short paper. We point out that it is not uncommon for similar works to include only one language pair (e.g. one of the suggested papers (Chen et al., WMT 2022) also only considers one language pair). As such, we leave investigation of different language pairs and domains to future work.
>
> References:
>
> [Robust MT Evaluation with Sentence-level Multilingual Augmentation](https://aclanthology.org/2022.wmt-1.43) (Alves et al., WMT 2022)
>
> [Exploring Robustness of Machine Translation Metrics: A Study of Twenty-Two Automatic Metrics in the WMT22 Metric Task](https://aclanthology.org/2022.wmt-1.46) (Chen et al., WMT 2022)

---

### Meta-Review · Senior_Area_Chairs · 2023-10-04

**Recommendation:** 3

**Metareview:**

This paper investigates the robustness of machine translation (MT) metrics in the face of adversarial attacks. The primary contributions are the exploration of adversarial attacks that do not change translation semantics and the evaluation of MT metrics' responses to these attacks, particularly focusing on BERTScore, BLEURT, and COMET.

Based on the average scores, the paper is considered sound and has potential, but it is not without weaknesses. It falls short in terms of excitement, mainly due to perceived incremental novelty and the need for revisions.
In conclusion, while the paper addresses an important issue in the field of MT metrics, it should strive to provide stronger evidence for its claims, highlight its unique contributions, consider expanding the scope, and address the reviewers' concerns for a higher chance of acceptance.

---

### Decision · Program_Chairs · 2023-10-07

**Decision:**

Accept-Findings

**Comment:**

This paper investigates the robustness of machine translation (MT) metrics in the face of adversarial attacks. The primary contributions are the exploration of adversarial attacks that do not change translation semantics and the evaluation of MT metrics' responses to these attacks, particularly focusing on BERTScore, BLEURT, and COMET.

Based on the average scores, the paper is considered sound and has potential, but it is not without weaknesses. It falls short in terms of excitement, mainly due to perceived incremental novelty and the need for revisions.
In conclusion, while the paper addresses an important issue in the field of MT metrics, it should strive to provide stronger evidence for its claims, highlight its unique contributions, consider expanding the scope, and address the reviewers' concerns for a higher chance of acceptance.